# Comparing the impact of vaccination strategies on the spread of COVID-19, including a novel household-targeted vaccination strategy

André Voigt[1]*, Stig Omholt[2], Eivind Almaas[1,3]

**1** Department of Biotechnology and Food Science, NTNU - Norwegian University of Science and Technology, Trondheim, Norway, **2** Department of Circulation and Medical Imaging, NTNU - Norwegian University of Science and Technology, Trondheim, Norway, **3** K.G. Jebsen Center for Genetic Epidemiology, NTNU - Norwegian University of Science and Technology, Trondheim, Norway

* andre.voigt@ntnu.no

**Data Availability Statement:** All data is available at Figshare, under dx.doi.org/10.6084/m9.figshare.15178386.

## Abstract

With limited availability of vaccines, an efficient use of the limited supply of vaccines in order to achieve herd immunity will be an important tool to combat the wide-spread prevalence of COVID-19. Here, we compare a selection of strategies for vaccine distribution, including a novel targeted vaccination approach (EHR) that provides a noticeable increase in vaccine impact on disease spread compared to age-prioritized and random selection vaccination schemes. Using high-fidelity individual-based computer simulations with Oslo, Norway as an example, we find that for a community reproductive number in a setting where the base pre-vaccination reproduction number $R = 2.1$ without population immunity, the EHR method reaches herd immunity at 48% of the population vaccinated with 90% efficiency, whereas the common age-prioritized approach needs 89%, and a population-wide random selection approach requires 61%. We find that age-based strategies have a substantially weaker impact on epidemic spread and struggle to achieve herd immunity under the majority of conditions. Furthermore, the vaccination of minors is essential to achieving herd immunity, even for ideal vaccines providing 100% protection.

## Introduction

Population-wide vaccination is the most promising approach for long-term COVID-19 disease management [1]. Due to limited vaccine availability, in particular in the early stages of deployment, most authorities have to choose between vaccination prioritization schemes, aiming for a balance between total mortality, years of life lost, and cumulative levels of infections.

In responding to the COVID-19 pandemic, many countries have opted for a strategy of suppressing infection rates (with varying success) through stringent social, work and travel restrictions until a sufficient amount of vaccines are deployed to adequately mitigate the public health risks of COVID-19. There are two criteria which individually satisfy the latter objective.

**Funding:** AV, EA - Norwegian Research Council Grant 270068: ELIXIR Norway - a distributed infrastructure for the next generation of life science Norwegian Research Council: https://prosjektbanken.forskningsradet.no/en/explore/projects The funders had no role in study design, data collection and analysis, decision to publish, or preparation of the manuscript.

**Competing interests:** The authors have declared that no competing interests exist.

The first is achieving comprehensive vaccine coverage for those groups at a non-acceptable risk of death or severe illness. In this case, lifting restrictions may still lead to an increase in disease prevalence but with limited adverse public health effects. The second is by sufficiently inhibiting transmission that the reproductive number of the disease drops below unity, preventing exponential growth (also known as herd immunity), thus protecting vulnerable individuals from illness by limiting their risk of exposure.

In most European countries and the US, health care workers and the elderly were initially prioritized to receive vaccinations in order to reduce mortality and maintain health service readiness [2, 3]. However, in areas where prevalence is very low, any reduction in mortality rate would translate into limited gains in terms of prevented deaths. In such conditions, a strategy that more effectively targets case prevention may be a more efficient approach, even if it has a lesser impact in terms of per-case risk of serious illness and/or death. Where this is the case, as vaccines not only prevent infection but also may significantly inhibit transmission [4], one should identify vaccination strategies that bring the reproductive number $R$ below unity with the least vaccination effort, as this would enable stable long-term suppression of cases to near zero.

The main objective of this article is to compare the effect of different vaccination strategies on epidemic spread and their ability to achieve herd immunity, even as disease control measures are lifted. As the exact effect on social contact (and consequently R) of lifting current measures is unknown, we opt for an incremental approach, where we gradually increase the amount of contact in the model's random layer, simulating increased activity as a result of lifting of restrictions. By varying the amount of random contact, we can therefore create a set of baselines, for which we can compute $R_0$ assuming there is no immunity in the population. For each baseline, we can then produce a set of vaccine distributions defined by the amount of individuals fully vaccinated (as a fraction of the population) and the strategy used to select individuals for vaccination.

Multiple variants of the SARS-CoV-2 virus are currently in circulation world wide. The initial Wuhan strain, for which the vaccines were developed, has a markedly lower potential for transmission than several later strains. During spring and summer of 2021, the two strains that dominate global infections are the B.1.1.7 (Alpha) and B.1.617.2 (Delta) strains. The Alpha strain is estimated to be 40–80% more transmissible than the original strain [5–7] that spread world-wide, whereas the Delta strain is considered to be 40–60% more transmissible than the Alpha strain. In this study, we report results for both Alpha and Delta, since their transmission properties may affect the efficacy of vaccination strategies that intend to curb spread.

## Materials and methods

### Computational model

The results in this article were obtained through the use of a multi-layer individual based epidemic model, previously described in [8]. Briefly, the model represents daily spread and progress of COVID-19 for a city with demographics similar to Oslo, Norway, with approx. 700,000 inhabitants. Each inhabitant is represented as a discrete entity defined by personal attributes (age, COVID-19 state, and history). Each individual is then assigned to one or more cliques, through which it comes into contact with other individuals. These cliques correspond to interactions which are repeated over an extended period of time, such as households, workplace contact among colleagues, schools, child daycare and elder care facilities. In addition to these cliques, for each new day, each individual also comes into contact with a random group of individuals chosen from the entirety of the municipal population, according to the following rules:

- Each individual is assigned a maximum activity level, representing the average relative amount of daily random contact for that individual.

- For working-age adults (18 to 67 years old), the maximum activity level is given by the the sum of two random variables, from a normal distribution (with a mean of 10 and a standard deviation of 3) and a power-law distribution (given as $y = x^{-0.5}$, 100 with $x$ being a uniform random number in [0, 1]) respectively. The power-law factor is intended to replicate the concept of "super-spreaders", according to which certain socially central individuals have a substantially higher than average propensity for spread. Activity levels for minors and retirement-age adults do not include the power-law component.

- Each day, an individual's daily amount of random contact is randomly drawn from a uniform distribution ranging from 0 to that individual's maximum activity level.

The disease dynamics in the model follow a typical extended SEIR model, in which any individual can be either susceptible, exposed, infected asymptomatic, infected pre-symptomatic, infected symptomatic, hospitalized or dead [8]. When a susceptible individual comes into contact with an infected person, there is a layer-dependent probability that the susceptible individual transitions to the exposed state. An exposed individual will, after a period of time, progress to either an asymptomatic or a pre-symptomatic state. The first state corresponds to those infections for which symptoms never manifest and will transition next to the recovered state. Pre-symptomatic individuals will at some point develop symptoms, with a further possibility of hospitalization and/or death. As an individual transitions to any state except recovered or dead, the time spent in the new state as well as the following transition to be made is chosen at random according to a parameter table described in Ref. [8]. Recovery is assumed to provide immunity to further re-infection for at least the duration of the simulation.

For a given parameter set (disease infectiousness, disease control measures, and vaccinated population), epidemic spread on the model will follow a steady-state exponential growth described by a constant reproduction number $R$ as long as infection-induced immunity remains at a low level. Due to the multi-layer dynamics involved, a sudden change in parameters will lead to a transitional period of about 10 days until $R$ stabilizes at the steady-state for the new parameter set, as described in Ref. [8] (Supplementary material).

## Simulation process and computing $R$

In order to estimate $R$ for a given combination of social contact and vaccine distribution, we perform simulations in a three-step protocol intended to represent COVID-19's ability to spread from a few localized clusters. The first step of model initialization consists of the following two tasks:

1. 20 randomly selected individuals are infected as seed cases. No individuals are vaccinated at this point.

2. Infection is allowed to spread from the seed cases at an accelerated rate (representing a combination of free local spread and case importation) until the system reaches 100 symptomatic cases. No vaccinations or other infection control measures are in place at this point.

As the first task infects all the seed cases at the same time, the number of cases may briefly dip or flatten out after about one generation (9 days) as most of the initial cases recover within a few days from each other (see S1 Fig for an illustration), in particular when $R$ is relatively low. This effect dissipates within two generations. The purpose of the second task is to allow time for this to happen.

The model is then adjusted to the desired amount of random contact, and the population is simultaneously and suddenly immunized according to the chosen vaccination strategy and capacity. Our simulations assume a hypothetical single-dose vaccine with a specified efficiency, which represents the proportion of vaccinated individuals effectively immunized by the vaccine.

This approach represents a substantial simplification of real-world vaccination deployments, which gradually take place and typically revolve around multiple types of vaccines with varying degrees of efficiency [9–11], and many of which only provide partial protection after a single dose and require a second dose, administered some time after the first, in order to provide maximal protection. Nonetheless, the modelling approach of simultaneous vaccination of a specified fraction of the population can be seen as representing a "snapshot" of a gradual deployment at a given point in time. Consequently, varying vaccination fractions for a given strategy can be interpreted as different time points in a gradual vaccination deployment.

With the specified fraction of the population vaccinated and amount of random contact set to the desired level, the model runs for 40 (in-model) days. We then compute an individual reproductive number $R$ for each person that has recovered before the 40-day mark by simply directly counting the number of secondary infections caused by that person over the course of their illness. For a given day, we define a daily $R$ as the average of individual reproductive numbers for all persons that are sick on that day (including secondary infections caused by each individual on other days). In reporting a final $R$ for a given combination of activity and vaccination, we take the average of daily R-values from days 12 to 17 of the 40 days in this second step.

The first 11-day interval serves to pass the transient phase between change of contact activities from the initialization step, mentioned in the last paragraph of the section titled "computational model" above. The 23 day-gap after day 17 serves to ensure that all persons infected up to day 17 have had time to recover, in order to get an accurate assessment of the number of secondary infections caused by all persons counted on that day. An insufficient gap would mean that the subset of infected persons that have time to recover (which are the only ones for which we can count an accurate individual reproduction number) would be biased towards those with shorter illnesses, who would have less time to cause secondary infections. This would cause an underestimation of the actual average $R$.

## Robustness and handling uncertainty in re-opening scenarios

As mentioned in the introduction, accurate estimates of what $R_0$ for COVID-19 would be for Oslo in the absence of general health measures is unknown. In particular, it is hard to quantify exactly how much the risks of exposure and infection in each layer would increase in a full reopening scenario. Especially the random layer is subject to uncertainty, as the model is primarily calibrated for the period of April to August 2020 when public health measures were mainly directed towards restrictions on leisure activities and travel. For each of the Alpha and Delta variants, we therefore compute $R$ for a variety of infection probabilities in the random layer. As the daily number of random contacts for any given individual is much lower than the municipal population, reducing the infection probability in the random layer is equivalent to and interchangeable with a reduction in the number of random contacts each individual makes. This approach can therefore be considered to represent a mix of contact-reducing measures (such as avoidance of public venues) as well as as infection-reducing measures for the contacts that still happen (such as mask wearing). The other layers are kept at fixed infection probabilities with no particular restrictions (with the exception of symptomatic individuals self-isolating), corresponding to the pre-lockdown regime described in [8]. The relative

proportion of random contacts is therefore variable across scenarios, with higher values of $R$ having a higher share of infections taking place between random contacts. This also means that $R_0$ is not something that we explicitly set in advance for a given simulation, but rather is something that follows from the combination of per-layer infection probabilities.

In addition to the random layer, one would also expect some uncertainty attached to the amount of contact in the non-random layers in the event of a full re-opening. The way the model is set up, specifying by-layer infection probabilities which are a combination of human behavior and pathogen properties, robustness analysis with respect to contact patterns in the non-random layers is equivalent to adjusting infection probabilities, as is done for the Alpha and Delta comparison.

## Vaccination strategies

For the purposes of this article, we consider four vaccination strategies: A random approach where all individuals are equally likely to be selected for vaccination (regardless of age or other characteristics), one where selection is according to decreasing age (the eldest are vaccinated first), one in which adults (18 and above) are selected by increasing age, and a novel strategy we call "effective household reduction" (EHR) which we describe below in more detail. The motivation for the EHR strategy comes from the observation that a substantial share of COVID-19 infections happen between individuals in the same household [12], and an observed association between the number of large households and the prevalence of COVID-19 [8, 13]. EHR aims to disrupt infection by targeting large households as possible infection hubs. In summary, EHR can be described as allocating each marginal vaccination by choosing the oldest individual from one of the households with the largest number of unvaccinated members.

As an example, consider a hypothetical community of 5 households labeled A through E, with 5 individuals in household A, 4 in household B, 3 in C, 2 in D, and 1 in E. This gives a total of 15 individuals, none of which are previously immune. Suppose that we only have the capacity to fully vaccinate 5 individuals. In line with the reasoning above, EHR gives the following vaccination strategy:

1. The largest household by effective size is A, with 5 individuals. The first vaccine is therefore allocated to the eldest member of A.

2. A and B are now tied in terms of effective size, with 4 susceptible individuals each. We therefore vaccinate one person each from both A and B.

3. A, B, and C are now tied in terms of effective size, with 3 individuals each. With enough capacity left to vaccinate only two people, we therefore randomly choose two of the three households, and vaccinate one individual in each of the randomly chosen households.

Key details for the Oslo model, namely household composition demographics and the resulting relationship with vaccine allocation for a given total vaccinated population, are provided in S2 and S3 Figs. Regarding point 3. above, one could introduce tie-breaking schemes, such as prioritizing households according to highest average age or the age of the oldest household member. However, as shown in S4 Fig, these seem to have a minimal impact on the effectiveness of the strategy.

## Materials

All code used for this article is available at dx.doi.org/10.6084/m9.figshare.15178386. Developments to the underlying model are available at https://github.com/andrevo/covid19-ntnu.

## Results

The central premise of our proposed vaccine strategy of "effective household reduction" (EHR), is that a household's effective size from an epidemic point of view is equal to the number of susceptible people in that household. This effective size is the actual number of people minus the number of immune people in said household, as immune individuals are essentially non-existent in the context of an epidemic (in the event of partial immunity, the effective size is reduced by the number of vaccinated individuals multiplied by the efficiency of the vaccine). It may be tempting to target household infections by vaccinating all individuals in the largest households, since an individual's ability to spread infection within their household increases with the number of susceptible individuals. However, as soon as a single individual is vaccinated, the effective household size falls by one when assuming ideal protection, thus reducing the relative benefit of vaccinating the remaining household members compared to the first individual.

Comparing the presented strategies (Fig 1), we see that for both the COVID-19 Alpha and Delta variants, EHR consistently outperforms the other three strategies with regards to reduction of spread (in our scenario assuming $R_0 = 3$), random vaccination comes second, followed by vaccination by increasing age and vaccination by decreasing age coming last. Here, we will use the notation $R(S, F, E)$ to denote the reproductive number in a population with with a fraction $F$ vaccinated according to strategy $S$. For instance, $R(EHR, 20\%, 90\%)$ is the reproductive number in a population with 20% of the population vaccinated with 90% efficiency according to the EHR strategy. Comparing EHR to the other strategies, we find that the gap increases with the vaccinated fraction of the population until about 50%, with $R(EHR, 50\%, 100\%) = 1.34$ vs $R(Random, 50\%, 100\%) = 1.62$, with slightly smaller gaps at 90% and 70% efficiency (in the case of the Delta variant), as reported in Table 1. As people get vaccinated past this point, the difference begins to narrow, overlapping (within error bars) at 100% vaccination coverage (which should trivially be true, as there is only one way to select 100% of the population). We also note that vaccinating children appears to be an essential requirement in achieving herd immunity, with the adult-only strategy failing to achieve herd immunity even for ideal vaccines providing 100% protection. Interestingly, with 70% vaccine efficiency, none of the strategies achieve herd immunity even at 100% vaccination coverage. Looking at the intersect between the R-curves and the herd immunity threshold (i.e., $R = 1$), we find that EHR exhibits a modest advantage compared to random vaccination, with a more substantial edge relative to the elderly-first strategy.

The actual $R_0$ for COVID-19 is still subject to substantial uncertainty. Looking at a variety of re-opening scenarios (Fig 2), we find that EHR and random vaccination perform quite similarly (with a minor edge in the favor of EHR, as seen in Fig 1), but substantially better than either of the age-based strategies. For lower values of $R_0$, EHR achieves herd immunity faster than the random strategy, but this gap narrows as $R_0$ increases.

Further investigating herd immunity thresholds for a range of values for $R_0$ (Fig 3, Table 2), we find that for most parameter choices, EHR achieves herd immunity with fewer vaccinated individuals than the other presented strategies (and universally, dramatically earlier than age-based strategies). At the most extreme, assuming a 90% effective vaccine and $R_0 = 2$, EHR achieves herd immunity at just 48% of the population vaccinated, compared to thresholds of 61% and 89% for random and eldest-first strategies. At 100% efficiency, the relative herd immunity thresholds are 44% and 55%, for the EHR and random strategies, respectively.

The difference narrows as $R_0$ increases, or with less effective vaccines. At 70% vaccine efficiency, the herd immunity fraction for EHR and random strategies are close to indistinguishable (within error bars) across all values for $R_0$. At 90% effectiveness, the intersection (within

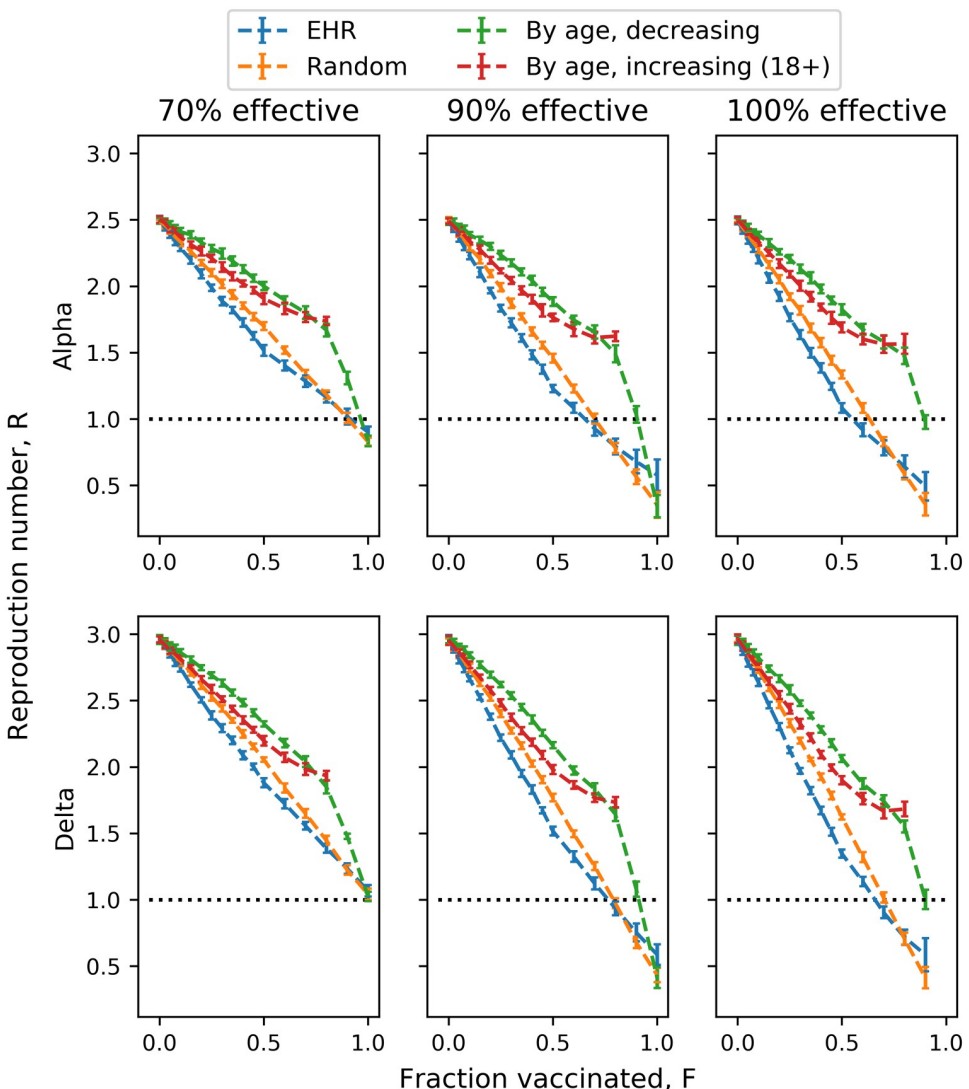

**Fig 1. Effect of vaccine deployment on the reproductive number, according to strategy and variant.** Epidemic reproduction number as a function of vaccinated fraction of population and vaccine efficiency for each of the four presented strategies, assuming a base reproduction number of $R_0 = 2.5$ for the COVID-19 Alpha variant and $R_0 = 3$ for Delta. Each data point (mean and standard error bar) is based on 20 duplicate simulations.

**Table 1. Effect of vaccine deployment on $R$, the reproductive number, in response to vaccination strategies for COVID-19 Alpha and Delta variants.** Epidemic reproduction number as a function of vaccinated fraction of population and vaccine efficiency for each of the four presented strategies, assuming a base reproduction number of $R_0 = 2.5$ for the COVID-19 Alpha variant and $R_0 = 3$ for the Delta variant. Each data point (mean and standard error) is based on 20 simulations.

| | | Reproductive number $R$ | | | | | |
| | | 70% effective | | | 90% effective | | |
| | Vacc. | Random | EHR | Decr. age | Random | EHR | Decr. age |
|---|---|---|---|---|---|---|---|
| Alpha | 20% | 2.10 ± 0.03 | 1.99 ± .02 | 2.28 ± .03 | 1.99 ± .03 | 1.83 ± .02 | 2.23 ± .03 |
| | 50% | 1.70 ± .03 | 1.52 ± .04 | 2.00 ± .03 | 1.46 ± .03 | 1.23 ± .03 | 1.88 ± .03 |
| | 80% | 1.18 ± .03 | 1.16 ± .04 | 1.67 ± .05 | 0.78 ± .04 | 0.79 ± .06 | 1.49 ± .06 |
| Delta | 20% | 2.52 ± .03 | 2.39 ± .03 | 2.69 ± .02 | 2.40 ± .02 | 2.22 ± .03 | 2.61 ± .02 |
| | 50% | 2.05 ± .02 | 1.88 ± .03 | 2.32 ± .02 | 1.77 ± .03 | 1.52 ± .03 | 2.16 ± .02 |
| | 80% | 1.44 ± .03 | 1.39 ± .03 | 1.85 ± .05 | 0.97 ± .04 | 0.93 ± .05 | 1.64 ± .05 |

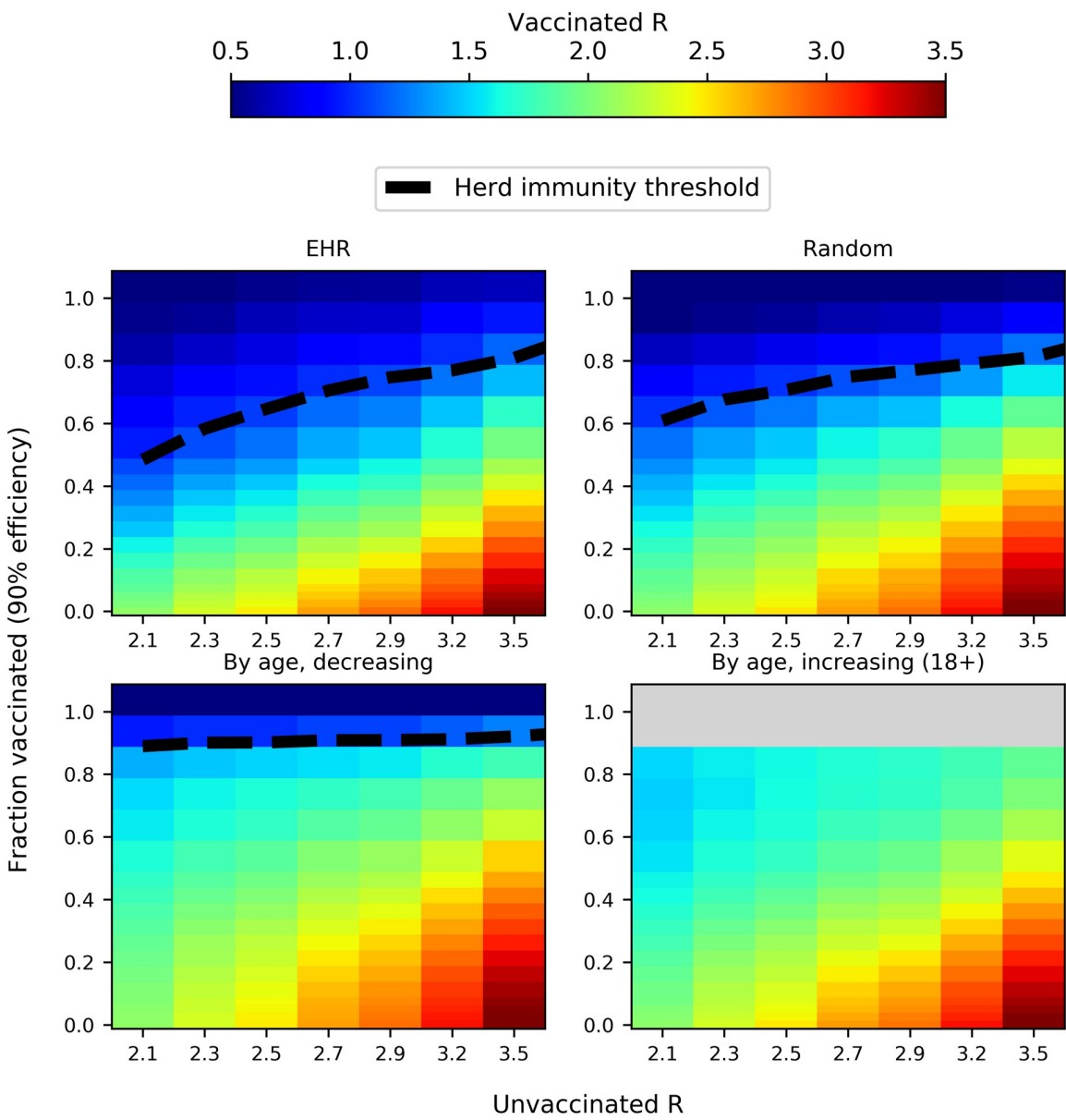

**Fig 2. Impact of different vaccination strategies on the reproductive number $R$ according to varying pre-vaccination $R_0$.** $R_0$ is adjusted by varying the amount of contact in the random layer of the model, while other layers (schools, workplaces, households) are kept constant. Dashed lines indicate the fraction of population which needs to be vaccinated in order to achieve herd immunity $R < 1$ depending on $R_0$ for an unvaccinated population. Herd immunity thresholds are also shown for random and age-dependent vaccination for comparison purposes. Grey denotes infeasible vaccination fractions ($>80\%$) for the 18+ strategy. The depicted $R$ values are averages of 20 duplicate simulations for each data point.

error bars) of the EHR and random curves happens for $R_0 \approx 3$, and at 100% efficiency, the intersect happens somewhere between $R_0 = 3.2$ and $R_0 = 3.6$.

## Discussion

In general, we find that EHR performs noticeably better than other strategies for lower values of $R_0$, while it performs comparable to random vaccination (but still dramatically better than age-based strategies) for higher values of $R_0$. While the actual value of $R_0$ for COVID-19 is

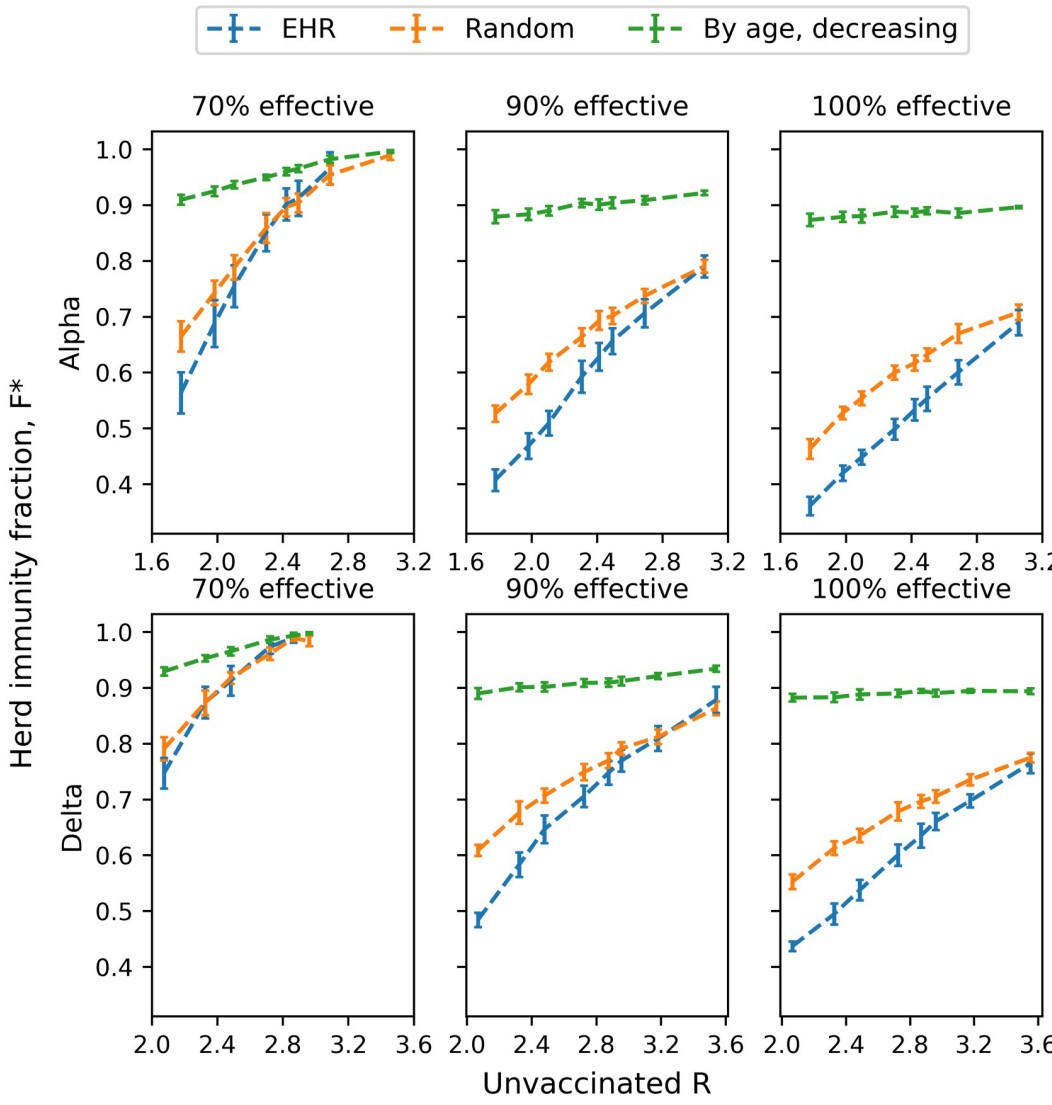

**Fig 3. Herd immunity thresholds according to $R_0$ and choice of strategy.** Fraction of population requiring vaccination in order to reach herd immunity ($R < 1$) according to each strategy and $R_0$ for an unvaccinated population. Each data point and corresponding errorbar represent the mean $R$ and standard deviation of 20 duplicate simulations, respectively.

**Table 2. Herd immunity thresholds $F^*$ in response to $R_0$ and choice of strategy according to strain.** Fraction of population requiring vaccination in order to reach herd immunity ($R < 1$) according to each strategy and $R_0$ for an unvaccinated population. Each data point (mean $R$ and standard error) is based on 20 simulations.

| | | Herd immunity threshold, $F^*$ | | | | | |
| | | 70% effective | | | 90% effective | | |
| | $R_0$ | Random | EHR | Decr. age | Random | EHR | Decr. age |
|---|---|---|---|---|---|---|---|
| Alpha | 1.8 | 66.4 ± 2.7% | 56.3 ± 3.7% | 90.9 ± 0.9% | 52.6 ± 1.5% | 40.7 ± 1.9% | 87.9 ± 1.1% |
| | 2.4 | 89.6 ± 1.7% | 90.1 ± 2.9% | 96.0 ± 0.6% | 69.3 ± 1.7% | 62.8 ± 2.5% | 90.1 ± 0.9% |
| | 3.1 | 98.9 ± 0.8% | >100% | 99.6 ± 0.3% | 79.0 ± 1.1% | 79.0 ± 2.0% | 92.2 ± 0.4% |
| Delta | 2.1 | 79.1 ± 2.1% | 74.7 ± 2.8% | 92.9 ± 0.7% | 60.8 ± 1.0% | 48.4 ± 1.3% | 89.0 ± 1.0% |
| | 2.7 | 96.1% ± 1.1 | 97.3 ± 1.3% | 98.6 ± 0.6% | 74.9 ± 1.5% | 70.5 ± 1.9% | 90.9 ± 0.7% |
| | 3.5 | >100% | >100% | >100% | 86.3 ± 1.2% | 87.8 ± 2.4% | 93.4 ± 0.5% |

unknown, it can reasonably be expected to be higher than 3, particularly for Delta and other recent variants. In this event, the necessary vaccination coverage for random and EHR strategies in order to lift all social restrictions is likely to be close to equal. Nonetheless, intermediate values of $R_0$ (between 2 and 3) in our model can be interpreted as reflecting partial lifting of general measures on a population-wide level (i.e., measures which affect the population evenly). Both for reasons of economic and social costs as well as due to inherent uncertainties in $R_0$, maintaining a maximally strict set of restrictions throughout the vaccination roll-out, which is then suddenly lifted once a critical pre-determined vaccination threshold has been reached, is a difficult proposition. If one instead opts for a gradual lifting of measures as the vaccination coverage increases, an EHR strategy should allow for a somewhat faster lifting of general restrictions as the deployment of vaccines progresses.

Much discussion has been made concerning the role of super-spreaders in the spread of COVID-19. In fact, the EHR strategy fundamentally rests on the notion, common in network-based approaches to studying epidemic spread and immunization [14–17], of targeting heterogeneities in how well individual nodes (i.e. persons) are able to spread the disease. If a network were perfectly homogeneous, there would be no preferential target for immunization; conversely, if only one node is able to infect others, targeting this node would be an exceptionally efficient strategy. In any targeted strategy, if we begin by immunizing those individuals with the highest spreading potential, the marginal impact of the first immunization is greater than the immunization of an average individual; however, as we move down the ranks one necessarily reaches the point at which there are no more higher-than-average spreading nodes to be immunized. Consequently, the more nodes a targeted strategy needs to immunize, the lesser the difference between it and a randomized strategy.

As such, the maximum possible relative benefit of a targeted strategy rests on two main factors: the amount of heterogeneity in spreading potential, and the necessary number of people that need to be immunized. The latter is partially dependent on the former, but also on factors such as the efficiency of immunization and the general infectiousness of the disease. While vaccines for COVID-19 provide excellent protection, the disease is also highly infectious. As we find that EHR still requires substantial vaccine coverage to achieve herd immunity, it is tempting to consider alternate vaccination strategies which might more precisely target the individuals responsible for a majority of infections. This is challenging for several reasons. Firstly, even if a factor in itself might be identified, such as certain co-morbidities or leisure activities, properly identifying which individuals actually belong to the corresponding group might be outside the capabilities of relevant authorities. Second, prioritizing vaccines according to certain types of adjustable behavior might be politically controversial, as it might be perceived as an incentive to (or reward for) engaging in such behavior. Third, super-spreading potential may also be driven by factors such as viral load [18, 19], which may be inherently unpredictable at the individual level, making it impossible to pre-determine which individuals are likely to cause the most secondary infections. With these concerns in mind, EHR has substantial advantages in that household sizes constitute a clearly definable criterion, can be identified within the capabilities of public authorities (as evidenced by available data from census and national statistics agencies in many countries [20–22]) and are also likely to be a less politically controversial target than more behavior-based alternatives.

While EHR does outperform the other strategies, the likely high reproductive number of current strains of COVID-19 requires extensive vaccination coverage in order to achieve herd immunity, enough that the choice of strategy has a reduced impact in terms of the herd immunity threshold. Even if surmountable, the organizational challenges of such a strategy may not justify the gains with regards to COVID-19. The benefits may be greater, however, in the event of a possible future pandemic with a lower $R$, but for which the mortality is high enough that

strict public health countermeasures need to be enforced until a vaccine is developed and deployed. For such a disease, herd immunity thresholds would be lower, increasing the viability of targeted strategies such as EHR.

The difficulty with which age-dependent strategies achieve herd immunity, even with highly effective vaccines, may seem surprising at first. There are several explanations for this. In our model, young adults and children tend to have a large amount of exposure, which increases their infectious potential (S5–S7 Figs). This is generally consistent with existing research [23, 24]. Social contact patterns in the model are also partially age-assortative(people tend to associate with people in their age group), particularly for minors [23, 25, 26]. While these have a certain amount of contact with adults (primarily family and educational staff), most of their contacts are with other minors, primarily schoolmates and household members. In a situation in which all (or nearly all) adults are vaccinated while all minors remain unvaccinated, any infected minor would still infect as many other minors as if no adults were vaccinated. In the case that this number is higher than one, it follows that one cannot achieve herd immunity through vaccination of adults alone, as long as the bulk of the population of minors remains susceptible to infection and is able to sustain exponential growth.

For adults, and to a lesser degree for children, we note that the observed age assortativiy in our model is actually substantially coarser than contact networks determined through survey-based studies [23, 25], as the Statistics Norway database did not provide detailed data for age compositions within households or workplaces (only number of members and qualitative data such as types of household), while the random layer is well-mixed due to computational constraints. This also means that model also exhibits less exposure between teenagers and young adults than what would be expected from the survey-based matrices, as one would expect most of these interactions would take place within the household (for which it is hard to quantify the proportion of young adults still living with their families) or in the random layer (for which more elaborate age-to-age mixing patterns are computationally infeasible given the current model framework).

An important difference between the contact matrices presented in [23–26]and S6 Fig lies in the fact that our model does not explicitly model one-to-one social encounters in high detail, but rather associates a general daily risk of transmission between two individuals according to common participation in some aspect of life. Empirical contact networks are not necessarily perfectly proportional with exposure risk, as the relationship between the distance and duration of a social contact on one hand and transmission risk on the other is likely to be complex, in addition to factors such as air circulation and differences in hygiene between different types of interpersonal contact. Instead, our model was calibrated by matching the proportions of infections in each layer with empirical Norwegian infection tracing data [8].

It is important to keep in mind that $R$ is measured as an average only over those individuals that actually get infected. As such, adults who would have very little spreading potential even if they were infected do not bring down the average to a great extent, as they make up a smaller portion of the infected population. Consequently, starting from any unvaccinated scenario such that an average infected minor is likely to infect, on average, at least one other minor in addition to any adults they may infect, $R$ will remain above unity even with full vaccine coverage amongst the adult population, assuming all minors remain unvaccinated.

For all vaccination strategies evaluated in this article, our single scoring criterion has been how well each strategy prevents spread, without regard for its impact on mortality rates. It is a common observation that strategies which are highly effective at suppressing spread are not so effective at minimizing mortality, and vice versa [27–29]. In particular, the choice often falls between vaccinating younger individuals (typically young adults, but occasionally children),

who are less susceptible to the disease but are more likely to spread it, and vaccinating the elderly, which have less spreading potential but are more vulnerable once infected. Typically, however, the scoring criterion in a given comparison is the total number of deaths in a given time period or until an epidemic can be declared over. In this case, a spread-reducing strategy is still evaluated on whether it ultimately reduces mortality in the population, not purely on how well it reduces spread. While this is a straightforward scoring criterion, it does not account for those cases where public authorities, faced with a rising case count and a higher-than-acceptable mortality for all strategies, will still see themselves forced to reach for $R < 1$ through social restrictions. In such an event, it is reasonable to expect that spread-reducing strategies would require less severe and less costly public health measures. If social restrictions achieve $R < 1$ while prevalence is still low, it also follows that the number of deaths will be low as long as restrictions are effective.

Our approach is particularly oriented towards situations where other measures are sufficient to keep the prevalence, and thus the number of deaths, at a negligible level until enough of the population is vaccinated. When the societal costs of maintaining such measures are high, it is essential to reach this point in the shortest possible time (typically meaning, with the fewest doses) possible. The low prevalence of disease in such a case means that the contrast between strategies in terms of lives lost remains minimal.

It is conceivable that a mortality-focused strategy (such as eldest-first) might achieve an acceptable mortality rate quicker than spread-focused strategies can achieve herd immunity, and would therefore allow for relieving disease control measures faster. In this case, a mortality-focused strategy would also be a preferable approach even if prevalence is low enough during vaccine deployment for the total number of deaths and severe illnesses to not be of particular concern. However, unlike the criterion of herd immunity, what constitutes a threshold for an acceptable mortality rate is fundamentally a moral and political question, rather than a scientific one; therefore, any attempt to evaluate how efficiently any given strategy would achieve acceptable mortality is outside the scope of this article.

## Conclusion

As mentioned in the introduction, vaccination for COVID-19 would allow for relieving disease prevention measures on the completion of at least one of two objectives: reduction in the per-case risk of serious illness and death to a level which is acceptable even with exponential spread of infection, or by achieving herd immunity through breaking of infection chains. In our simulations, we find that these two objectives conflict to some extent, and that a vaccination strategy focusing on the first objective by prioritizing vaccines according to descending age performs poorly with regards to the second objective.

The choice of vaccination strategy must be made on the basis of reducing total health and social costs over the duration of the epidemic. A strategy that requires more time in order to reach the objective might still be more appropriate if deaths and serious illness during this period are significantly reduced. A comprehensive cost-benefit evaluation of possible strategies involves, among other factors, weighing health costs against the time and resources required to implement it, and involves both ethical and scientific considerations.

However, in areas where other public health measures have been and remain sufficient for suppressing prevalence to a negligible or even non-existent level (such as parts of East Asia and Oceania), and where a low prevalence can be maintained until the vaccination program achieves a satisfactory coverage of the population, any possible direct health benefits from a mortality-focused strategy become minimal as long as other measures remain effective. In such cases, the ultimate purpose of the vaccination program effectively consists in relieving the

burden imposed by these other measures, for which an infection-targeted strategy such as EHR might be more suitable than mortality-targeted strategies such as eldest-first.

## Supporting information

**S1 Fig. Example initial epidemic progression for 5 replicate runs after seeding 100 individuals, using $R$ = 2.** Each run corresponds to a similar initialization with different random seeds. While individual curves evolve stochastically, each replicate is characterised by an steep rise from around 3-4 days into the simulation (corresponding to the incubation time), which slows down to a more moderate exponential trend from about 10 days from the start of the simulation.
(PNG)

**S2 Fig. Distribution of household sizes for the Oslo model.** Bars represent the amount of households in the computational model with the number of members indicated on the x-axis.
(PNG)

**S3 Fig. Relative share of vaccines cumulatively allocated to households of a given size as a function of total vaccinated population according to the EHR strategy.** Illustrating the EHR strategy on our Oslo model, we begin by allocating one vaccine to each of the ≈100 households of size 8. This means that all of the first ≈100 individuals vaccinated are members of 8-person households. The second stage in deployment consists of allocating the next ≈350 doses evenly (one each) to each of the ≈100 households of size 8 and ≈250 households of size 7, until a total of ≈450 individuals are vaccinated. In the third stage (individuals ≈450 through ≈6700), one more vaccine is allocated to each household of size 6 (≈5800 households), 7 or 8. This is repeated until all individuals are vaccinated, with more and more households included in each stage (by order of decreasing size). While members of large households make up the entirety of the first people vaccinated, their relative share of the vaccinated population drops as vaccine deployment progresses.
(PNG)

**S4 Fig. Impact on epidemic spread of different tie-breaking strategies for same-sized households using the EHR strategy.** Computationally determined reproduction numbers according to vaccinated fraction of population and choice of tie-breaking strategy for same-sized households in the EHR strategy, assuming the Delta variant with a pre-vaccine reproduction number of $R_0$ = 3. Random tiebreak and average household age are approximately equivalent, while priorizing households with older members performs slightly worse.
(PNG)

**S5 Fig. Age distribution for the Oslo population according to the model and reference data from Statistics Norway.** Number of Oslo inhabitants of a given age in the procedurally generated model, compared with real-life data provided by Statistics Norway (SSB), illustrating a generally close correlation.
(PNG)

**S6 Fig. Age-to-age exposure matrices resulting from clique composition and specified per-clique infection probabilities.** Each pixel indicates average the daily infection probability for each individual of age X resulting from one asymptomatic individual of age Y (assuming a regime with the Alpha variant at $R$ = 3.2). This probability consists of the sum across all layers of shared clique memberships between individuals of age X and age Y multiplied with the daily probability of infection for the corresponding layer, divided by the the product of the number of individuals of ages X and Y, resulting in a symmetrical matrix. While similar, this is

not strictly a contact matrix as in [23–26], as these contact matrices do not differentiate between the probability of transmission for different types of contact.
(PNG)

**S7 Fig. Relative infection risk according to age, normalized to peak infectious age (12 years).** Relative risk is defined as the average expected number of daily infections caused by an asymptomatic individual (assuming a regime with the Alpha variant at $R = 3.2$) in an entirely susceptible population, divided by the corresponding value for an asymptomatic 12-year old (0.29). Due to the symmetrical nature of contacts, this can also be interpreted as the relative daily risk of a susceptible individual of a given age becoming infected, assuming a uniform proportion of infectious individuals for all ages.
(PNG)

## Acknowledgments

We thank HUNT cloud (https://www.ntnu.edu/mh/huntcloud) for computing resources.

## Author Contributions

**Conceptualization:** André Voigt, Stig Omholt, Eivind Almaas.

**Data curation:** André Voigt.

**Formal analysis:** André Voigt.

**Funding acquisition:** Eivind Almaas.

**Investigation:** André Voigt.

**Methodology:** André Voigt, Eivind Almaas.

**Project administration:** Stig Omholt, Eivind Almaas.

**Resources:** André Voigt, Eivind Almaas.

**Software:** André Voigt.

**Supervision:** Stig Omholt, Eivind Almaas.

**Validation:** André Voigt.

**Visualization:** André Voigt.

**Writing – original draft:** André Voigt, Eivind Almaas.

**Writing – review & editing:** André Voigt, Stig Omholt, Eivind Almaas.

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
