## [Decision Letter · Decision Letter 0]

5 Oct 2021

PONE-D-21-27268Comparing the impact of vaccination strategies on the spread of COVID-19, including a novel household-targeted vaccination strategyPLOS ONE

Dear Dr. Voigt,

Thank you for submitting your manuscript to PLOS ONE. After careful consideration, we feel that it has merit but does not fully meet PLOS ONE’s publication criteria as it currently stands. Therefore, we invite you to submit a revised version of the manuscript that addresses the points raised during the review process.

The two thorough reports received agree that the manuscript is interesting and well written but still needs some clarification on the methodology and description of the model as well as to improve the discussion to help the reader and strengthen the drawn conclusions. In case you consider you can overcome all the criticisms and address the comments and suggestions made by the two reviewers, please prepare a revised version of your manuscript and a point by point rebuttal letter.  

We look forward to receiving your revised manuscript.

Kind regards,

Irene Sendiña-Nadal

Academic Editor

PLOS ONE

Journal Requirements:

2. Please note that PLOS ONE has specific guidelines on software sharing (http://journals.plos.org/plosone/s/materials-and-software-sharing#loc-sharing-software) for manuscripts whose main purpose is the description of a new software or software package. In this case, new software must conform to the Open Source Definition (https://opensource.org/docs/osd) and be deposited in an open software archive. Please see http://journals.plos.org/plosone/s/materials-and-software-sharing#loc-depositing-software for more information on depositing your software.

4. We note you have included a table to which you do not refer in the text of your manuscript. Please ensure that you refer to Table 2 in your text; if accepted, production will need this reference to link the reader to the Table.

Reviewers' comments:

Reviewer's Responses to Questions

**Comments to the Author**

1. Is the manuscript technically sound, and do the data support the conclusions?

Reviewer #1: Yes

Reviewer #2: Yes

2. Has the statistical analysis been performed appropriately and rigorously? 

Reviewer #1: Yes

Reviewer #2: N/A

3. Have the authors made all data underlying the findings in their manuscript fully available?

Reviewer #1: Yes

Reviewer #2: Yes

4. Is the manuscript presented in an intelligible fashion and written in standard English?

Reviewer #1: Yes

Reviewer #2: Yes

5. Review Comments to the Author

Reviewer #1: See attached pdf. See attached pdf. See attached pdf. See attached pdf. See attached pdf. See attached pdf. See attached pdf. See attached pdf. See attached pdf. See attached pdf. See attached pdf. See attached pdf.

Reviewer #2: The authors here leverage an agent based model introduced in a previous publication to characterize COVID-19 evolution in Oslo and explore the outcome of different vaccine prioritization strategies. Within the different choices to judge the suitability of a given strategy, the authors quantify how the vaccination of different groups shapes the effective reproductive number of the disease. Specifically, they find that the random vaccination of the population seems more efficient for that purpose than following age-based strategies. Moreover, they propose another strategy, the effective household reduction (EHR), which focus on targeting those individuals belonging to largely populated households. Overall, this strategy leads to a higher decrease of the effective reproduction number and therefore less vaccination efforts to reach herd immunity in the population.

The results presented in the manuscript are novel, the paper is scientifically sound and correct and the claims made by the authors are supported by the information shown in the figures and tables. Therefore, I think that the paper meets the criteria for publication in PLOS One. Nonetheless, prior to an eventual publication, I have some suggestions to improve the readability of the manuscript and to better contextualize the findings made by the authors based on the literature recently published addressing the same problem.

In this sense, I think that the debate raised by the authors about the conflict between prioritizing the disruption of the transmission chains (and therefore reducing the effective reproduction number) and other strategies targeting the most vulnerable population to reduce the death toll taken by the disease would benefit from including more references. For instance, I encourage the authors to discuss the similarities and differences of their findings with respect to the results reported in [K. Bubar et al. Science 371, 6532 (2021)] or in [J. Buckner et at. PNAS, 118 (26), E2025786118 (2021)].

The authors should extend the discussion on the results obtained in the manuscript. In this sense, I think that including some statistics on the number of interactions of each age group in the network of contacts would help the reader understand the roots of the failure of age-based strategies. It would be also very convenient to explain the origin of the crossover points between the random and EHR vaccination strategies observed in some scenarios. Can it be related to the fact that, when the pathogen infectiousness is high, it is more convenient to distribute vaccine across different affected households rather than targeting those with more members?

I think that the distinction between Alpha and Delta variants is a bit misleading because the intervals assumed for the basic reproduction number of both variants overlap each other. Given the fact that there is no clear agreement on the specific values associated with each variant, I would avoid making this classification and just introduce the analysis of the results in general terms, addressing how the outcome of the vaccination varies as a function of the basic reproduction number.

The authors should provide more details concerning the number of realizations carried out to obtain the different results shown throughout the manuscript. Likewise, the results presented in the different tables should include the confidence intervals extracted from the simulations to improve the statistical analysis.

As a minor detail, I think that there is a typo in Table 2 regarding the herd immunity value estimated for a randomly distributed vaccine with 90% efficacy and $R_0=3.1$

6. PLOS authors have the option to publish the peer review history of their article (what does this mean?). If published, this will include your full peer review and any attached files.

Reviewer #1: **Yes: **Claus Kadelka

Reviewer #2: No

---

## [Author Response · Author response to Decision Letter 0]

17 Nov 2021

See attached PDF titled "Response to Reviewers"

---

## [Decision Letter · Decision Letter 1]

7 Dec 2021

PONE-D-21-27268R1Comparing the impact of vaccination strategies on the spread of COVID-19, including a novel household-targeted vaccination strategyPLOS ONE

Dear Dr. Voigt,

Thank you for submitting your manuscript to PLOS ONE. After careful consideration, we feel that it has merit but does not fully meet PLOS ONE’s publication criteria as it currently stands. Therefore, we invite you to submit a revised version of the manuscript that addresses the points raised during the review process. In particular, please, consider the minor comments raised by the Reviewer #1. 

We look forward to receiving your revised manuscript.

Kind regards,

Irene Sendiña-Nadal

Academic Editor

PLOS ONE

Journal Requirements:

Reviewers' comments:

Reviewer's Responses to Questions

**Comments to the Author**

1. If the authors have adequately addressed your comments raised in a previous round of review and you feel that this manuscript is now acceptable for publication, you may indicate that here to bypass the “Comments to the Author” section, enter your conflict of interest statement in the “Confidential to Editor” section, and submit your "Accept" recommendation.

Reviewer #1: (No Response)

Reviewer #2: All comments have been addressed

2. Is the manuscript technically sound, and do the data support the conclusions?

Reviewer #1: Yes

Reviewer #2: Yes

3. Has the statistical analysis been performed appropriately and rigorously? 

Reviewer #1: Yes

Reviewer #2: Yes

4. Have the authors made all data underlying the findings in their manuscript fully available?

Reviewer #1: Yes

Reviewer #2: Yes

5. Is the manuscript presented in an intelligible fashion and written in standard English?

Reviewer #1: Yes

Reviewer #2: Yes

6. Review Comments to the Author

Reviewer #1: I want to thank the authors for adding missing details regarding the methodology. As a result, the manuscript is much improved and is in my opinion, with a few minor changes, suitable for publication in PLOS One.

Minor changes:

1. I appreciate the added explanatory supplementary figures. It would be helpful if the authors could add full figure legends (not just a caption) for each one of these figures, in particular the more complicated ones such as SFig 6.

2. The additional description of the random contact network (lines 62-71) helps but is still incomplete. It should always be the goal that, at least in theory, one can reproduce the results. This is only possible if the authors mention somewhere the used parameter values, e.g. for the normal and for the power-law distribution, and also what proportion of contacts are considered random contacts.

Other comments:

1. I appreciate the additional text (line 325-339). It helps to understand why the authors chose their approach. However, I still believe this study could benefit from inclusion of POLYMOD data, at least for the random contact level, which should in theory not be hard. One example where the author’s approach leads to clearly non-realistic contact patterns is the amount of contact between 14-18 and 20-24 year olds. SFig 6 shows that there are virtually no contacts between these two close age groups. This must be an artifact of the social clique approach.

Reviewer #2: (No Response)

7. PLOS authors have the option to publish the peer review history of their article (what does this mean?). If published, this will include your full peer review and any attached files.

Reviewer #1: **Yes: **Claus Kadelka

Reviewer #2: No

---

## [Author Response · Author response to Decision Letter 1]

10 Jan 2022

Response is provided in an attached letter.

---

## [Editor Report · Decision Letter 2]

13 Jan 2022

Comparing the impact of vaccination strategies on the spread of COVID-19, including a novel household-targeted vaccination strategy

PONE-D-21-27268R2

Dear Dr. Voigt,

We’re pleased to inform you that your manuscript has been judged scientifically suitable for publication and will be formally accepted for publication once it meets all outstanding technical requirements.

Kind regards,

Irene Sendiña-Nadal

Academic Editor

PLOS ONE
---

## [Editor Report · Acceptance letter]

24 Jan 2022

PONE-D-21-27268R2 

Comparing the impact of vaccination strategies on the spread of COVID-19, including a novel household-targeted vaccination strategy 

Dear Dr. Voigt:

I'm pleased to inform you that your manuscript has been deemed suitable for publication in PLOS ONE. Congratulations! Your manuscript is now with our production department. 

Kind regards, 

on behalf of

Dr. Irene Sendiña-Nadal 

Academic Editor

PLOS ONE